# Domain Adaptation Principal Component Analysis: Base Linear Method for Learning with Out-of-Distribution Data

**DOI:** 10.3390/e25010033

**Published:** 2022-12-24

**Authors:** Evgeny M. Mirkes, Jonathan Bac, Aziz Fouché, Sergey V. Stasenko, Andrei Zinovyev, Alexander N. Gorban

**Affiliations:** 1School of Computing and Mathematical Sciences, University of Leicester, Leicester LE1 7RH, UK; 2Institut Curie, PSL Research University, 75005 Paris, France; 3Institut National de la Santé et de la Recherche Médicale (INSERM), U900, 75012 Paris, France; 4CBIO-Centre for Computational Biology, Mines ParisTech, PSL Research University, 75005 Paris, France; 5Laboratory of Advanced Methods for High-Dimensional Data Analysis, Lobachevsky University, 603000 Nizhniy Novgorod, Russia

**Keywords:** principal component analysis, machine learning, domain adaptation, out-of-distribution generalization, transfer learning, single cell data analysis

## Abstract

Domain adaptation is a popular paradigm in modern machine learning which aims at tackling the problem of divergence (or shift) between the labeled training and validation datasets (source domain) and a potentially large unlabeled dataset (target domain). The task is to embed both datasets into a common space in which the source dataset is informative for training while the divergence between source and target is minimized. The most popular domain adaptation solutions are based on training neural networks that combine classification and adversarial learning modules, frequently making them both data-hungry and difficult to train. We present a method called Domain Adaptation Principal Component Analysis (DAPCA) that identifies a linear reduced data representation useful for solving the domain adaptation task. DAPCA algorithm introduces positive and negative weights between pairs of data points, and generalizes the supervised extension of principal component analysis. DAPCA is an iterative algorithm that solves a simple quadratic optimization problem at each iteration. The convergence of the algorithm is guaranteed, and the number of iterations is small in practice. We validate the suggested algorithm on previously proposed benchmarks for solving the domain adaptation task. We also show the benefit of using DAPCA in analyzing single-cell omics datasets in biomedical applications. Overall, DAPCA can serve as a practical preprocessing step in many machine learning applications leading to reduced dataset representations, taking into account possible divergence between source and target domains.

## 1. Introduction

The main and fundamental presumption of the traditional machine learning approach is that there is a probability distribution and that it is the same or very similar for the training and test sets. However, when the training set and the data that the model should use when operating are different, this assumption can be readily broken. The worst is that the new data lack known labels. Such situations are typical and lead to the problem of domain adaptation which became a popular challenge in modern machine learning [1,2,3,4].

The domain adaptation problem can be stated as follows. Let *S* be a labeled source dataset and *T* be an unlabeled target dataset, and let us further assume *S* and *T* are not sampled from the same probability distribution. The idea is to find a new representation of the data so that the non-labeled data would be as close to the labeled one as possible, with respect to the given classification problem (Figure 1).

This representation should be insensitive to the differences between the data distributions underlying source and target domains and, at the same time, should not hinder the classification task in the labeled source domain. The key question in domain adaptation-based learning is the definition of the objective functional: how to measure the difference between probability distributions of the source and the target domain sample. One possible approach consists of adversarial training [1,5]:Select a family of classifiers in data space;Choose the best classifier from this family for separating the source domain samples from the target ones;The error of this classifier is an objective function for maximization (large classification error means that the samples are indistinguishable by the selected family of classifiers).

In domain adaptations, one usually talks about two complementary subsystems that ideally must be trained simultaneously. The first one is a classifier that distinguishes the feature vector as either source or target and whose error is maximized. The second one is a feature generator that learns features that are as informative as possible for the classification task. Theoretical foundations of domain adaptation based on *H*-divergence between source and target domains and its estimates from finite datasets have been suggested in [5]. Here we understand domain adaptation as an approach to a more general out-of-distribution (OOD) generalization problem [6], and understand OOD as the situation where the unlabeled dataset has a distribution different from the labeled one.

One of the most popular applications of domain adaptation in computer vision was implemented using the framework of neural networks known as Domain Adaptation Neural Networks (DANN) [1,7], based on outlined above principle of combination of classification and adversarial learning modules. It is known that adversarial learning using neural networks is computationally heavy and data hungry. Therefore, it can be questioned if there exists a simple baseline linear or quasi-linear method for solving the supervised domain adaptation task which would be easier to compute with a small sample size. To the best of our knowledge, such a method has not been suggested so far. This situation is in contrast with other domains of machine learning where the baseline linear methods pre-existed in their generalizations using neural network-based tools (as trivial examples, linear regression pre-existed the sigmoidal multilayered perceptron and principal component analysis (PCA) pre-existed the neural network-based autoencoders).

The adversarial approach outlined above to reduce the shift between domains is not the only one that can be exploited for this purpose. Methods for aligning multidimensional data point clouds are well known in machine learning, and they can be used for solving domain adaptation tasks even without considering labels in the source domain. In particular, various generalizations of PCA or other matrix factorization approaches computing a joint linear representation of two and more datasets are widely exploited in machine learning [8,9,10,11]. Other linear methods such as Transfer Component Analysis minimizing the maximum mean discrepancy (MMD) distance [12] between linear projections of the source and the target datasets [3] and subspace alignment method [13] have been suggested. Correlation Alignment for Unsupervised Domain Adaptation (CORAL) aligns the original feature distributions of the source and target domains, rather than the bases of lower-dimensional subspaces and is claimed to be “frustratingly easy” but still effective in many applications approach to domain adaptation [14]. The computational simplicity of CORAL allows it to be introduced as a component of the loss function in training neural network-based classifiers and a deep transferrable data representation to be obtained [15]. The MMD measure can be also used for this purpose as in the Joint Adaptation Networks (JAN) framework where the joint maximum mean discrepancy (JMMD) criterion is optimized. A family of methods was suggested for searching such linear projections that are domain-invariant (i.e., mixing domains) and optimizing class compactness of data points projected from the source and the target domains [16]. This methodology uses labels in the source domain and introduces pseudo-labels in the target domain which was shown to be superior to TCA. Other methods based on computing the reciprocal data point neighborhood relations or application of optimal transport theory have become popular recently with many applications in various domains such as single-cell data science, with applications to data integration task [17,18].

In this study, we suggest a novel base linear method called Domain Adaptation Principal Component Analysis (DAPCA) for dealing with the problem of domain adaptation. It generalizes the Supervised PCA algorithm to the domain adaptation problem. The approach was first outlined in the context of one- and few-shot learning problems [19]. It relies on the definition of weights between pairs of data points, both in the source and the target domains and between them such that projections of data vectors onto the eigenvectors of a simple quadratic form would serve as good features with respect to domain adaptation. The number of such features is supposed to be smaller than the total number of variables in the data space: therefore, the method also represents a form of dimensionality reduction. The set of weights can depend on the features selected for representation: therefore, the base quadratic optimization method is accompanied by iterations such that at each iteration a simple quadratic optimization task is solved. As with many quasi-quadratic optimization iterative algorithms, convergence is guaranteed and, in practice, the number of iterations can be made relatively small.

There exist several linear domain adaptation methods, each of which is characterized by specific features: for example, some of them produce low-dimensional embedding of the source and target datasets, and some of them do not. A summary with a short description of their working principles is provided in Table 1.

## 2. Background

### 2.1. Principal Component Analysis with Weighted Pairs of Observations

Principal Component Analysis is one of the most used machine learning methods with applications in all domains of science (e.g., [22,23]). The classical formulation of the PCA problem belonged to Pearson and was introduced in 1901. It is based on the minimization of the mean squared distance from the data points to their projections on a hyperplane defined by an orthonormal vector base [20]. An alternative but equivalent (because of the Pythagorean theorem) definition of principal components is based on the maximization of the variance of projections on a hyperplane. This definition became the leading text book definition [24]. The third equivalent definition is the maximization of mean squared *pairwise* distance between the data points projections onto a hyperplane.

All these PCA definitions can lead to useful generalizations [25]. Generalization of the third above-mentioned definition by introducing weights for each pair of projections was explored in [21,26,27,28]. Below, we provide a short description of this method adapted to the purpose of this study.

Let us consider the standard PCA problem. Let a set of data vectors $x_{i}\in R^{d}$(i=1,…,N) be given, and let *P* be an orthogonal projector of $R^{d}$ on a *q*-dimensional plane. We search such a *q*-dimensional plane that maximizes the scattering of the data projections:(1)$$H=\frac{1}{2}\sum_{i,j=1}^{n}∥Px_{i}-Px_{j}{∥}^{2}=\frac{1}{2}\sum_{i,j=1}^{n}{∥P\left(x_{i}-x_{j}\right)∥}^{2}\to \max.$$For q=1, the scattering of projections (Equation 1) on a straight line with the normalised basis vector e is
(2)$$H=\frac{1}{2}\sum_{i,j=1}^{N}{\left(x_{i}-x_{j},e\right)}^{2}=N\left(\sum_{i=1}^{N}{\left(x_{i},e\right)}^{2}-{\left(\mu ,e\right)}^{2}\right)=N\left(e,Qe\right)$$
where μ is mean vector of the dataset *X*, the coefficients of the quadratic form (e,Qe) are the elements of the sample covariance matrix.

For an orthonormal basis $\left\{e_{1},\ldots ,e_{q}\right\}$ of the *q*-dimensional plane in data space, the maximum scattering of data projections (Equation 1) is achieved when $e_{1},\ldots ,e_{q}$ are the eigenvectors of *Q* corresponding to the *q* largest eigenvalues of *Q* (with taking into account possible multiplicity) $\lambda_{1}\geq \lambda_{2}\geq \ldots \geq \lambda_{q}$. This is precisely the standard PCA.

In practice, users are usually interested in solving an applied problem, such as classification or regression, rather than dimension reduction, which usually plays an auxiliary role. The first principal components might not align with the most informative from the classification point of view features. Therefore, ignoring a certain number of the first principal components has become a common practice in many applications. For example, the first principal components are frequently associated with technical artifacts in the analysis of omics datasets in bioinformatics, and removing them might improve the downstream analysis [29,30]. Sometimes it is necessary to remove more than ten first principal components to increase the signal/noise ratio [31].

Principal components can be significantly enriched in terms of the information they hold for the classification task if we modify the optimization problem (Equation 1) and include additional information in the principal component definition. One way of doing this is introducing a weight $W_{ij}$ for each pair of data points [19]:(3)$$H_{W}=\frac{1}{2}\sum_{i,j=1}^{n}W_{ij}{∥P\left(x_{i}-x_{j}\right)∥}^{2}\to \max.$$

It is reasonable to require symmetry of the weight matrix: $W_{ij}=W_{ji}$. Furthermore, we allow the weights $W_{ij}$ to be of any sign. As in the standard PCA, positive weights maximize the scattering of projections of data points on a hyperplane. In a sense, this can be viewed at as an effective repulsion of data point projections (obviously, the actual data points do not repulse in the actual data space). By contrast, negative weights try to minimize the distance between the corresponding pairs of data point projections, which can be viewed as an effective attraction of projections (see Figure 2).

Following the same logic as for (Equation 1), we consider the projection of (Equation 3) on a 1D subspace with the normalized basis vector e and define a new quadratic form with coefficients $q_{lm}^{W}$:(4)$$H_{W}=\sum_{lm}\left[\sum_{i}\left(\sum_{r}W_{ir}\right)x_{il}x_{im}-\sum_{ij}W_{ij}x_{il}x_{jm}\right]e_{l}e_{m}=\sum_{lm}q_{lm}^{W}e_{l}e_{m}.$$

For the *q*-dimensional planes the maximum of $H_{W}$ (Equation 4) is achieved when this plane is spanned by *q* eigenvectors of the matrix $Q^{W}=\left(q_{lm}^{W}\right)$ (Equation 4) that correspond to *q* largest eigenvalues of $Q^{W}$ (taking into account possible multiplicity) $\lambda_{1}\geq \lambda_{2}\geq \ldots \geq \lambda_{q}$ [19]. The difference compared to the standard PCA problem is that starting from some *q*, some eigenvalues can become negative, as clarified below.

There are several methods to assign weights in the matrix *W*:*Classical PCA*, $W_{ij}\equiv 1$;*Supervised PCA for classification tasks* [26,28]. Let us have a label $l_{i}$ for each data point $x_{i}$. The strategy to ‘attract the similar and repulse the dissimilar’ allows us to define weights as
(5)$$W_{ij}=\left\{\begin{array}{cc} 1 & \mathrm{if}l_{i}\neq l_{j}\left(\mathrm{repulsion}\right)\\ -\alpha & \mathrm{if}l_{i}=l_{j}\left(\mathrm{attraction}\right) \end{array}\right..$$*Supervised PCA for regression task*. In case the target attribute of data points is a set of real values $t=\{t_{1},\ldots ,t_{N}\}$, $t_{i}\in R^{1}$, the choice of weights in Supervised PCA can be adapted accordingly. Thus, we can require that projections of points with similar values of target attribute would have smaller weights, and those pairs of data points with very different target attribute values would have larger weights. One of the simplest choices of the weight matrix, in this case, is $W_{ij}={\left(t_{i}-t_{j}\right)}^{2}$.*Supervised PCA for any supervising task*. In principle, the weights $W_{ij}$ can be a function of any standard similarity measure between data points. The closer the desired outputs are, the smaller the weights should be. They can change the sign (from the repulsion of projections, $W_{ij}>0$ to the attraction, $W_{ij}<0$) or change the strength of projection repulsion.*Semi-supervised PCA* was defined for a mixture of labeled and unlabeled data [27]. In this case, different weights can be assigned to the different types of pairs of data points (both labeled in the same class, both labeled from different classes, one labeled and one unlabeled data point, both unlabeled). One of the simplest ideas here can be that projections of unlabeled data points effectively repulse (have positive weights), while the labeled and unlabeled projections do not interact (have zero weights).

The choice of the number of retained components for further analysis is a nontrivial question even for the classic PCA [32]. The most popular methods are based on evaluating the fraction of (un)explained variance or, equivalently, the mean squared error of the data approximation by the PCA hyperplane for different *q*. These methods take into account only the measure of approximation. However, in the case of Supervised PCA, the number of components needs to be optimized with respect to the final classification or regression task. For weighted PCA where some of the weights are negative, some of the eigenvalues can also become negative. Let us have *k* positive eigenvalues $\lambda_{1}\geq \lambda_{2}\geq \ldots \geq \lambda_{k}>0$ and d−k non-positive ones $0\geq \lambda_{k+1}\geq \ldots \geq \lambda_{d}$. Increasing the number of used principal components above *k* increases the accuracy of data set approximation but does not increase the value of the target function $H_{W}$ (Equation 4), so the data features defined by the principal components of order >k are not useful from the downstream classification task. Therefore, the standard practice is to use eigenvectors that correspond only to non-negative eigenvalues [33].

### 2.2. General Form of Supervised PCA for Classification Task

Let us consider the case of Supervised PCA for classification tasks (each data point $x_{i}$ has a discrete categorical label $l_{i}$). Let us denote as *n* the number of unique labels $L_{1},\ldots ,L_{n}$. We assign weights in such a way that in the space of projections on the first *q* principal components, the projection of points of the same class are effectively attracted, and the projections of points of different classes are repulsed. Therefore, we expect that the *q* first principal components will be more informative with respect to the downstream classification task than the standard principal components. The simplest weight definition, in this case, is (Equation 5), where α>0 is a parameter defining the “projection attraction force”.

This simplest weight definition can have undesired properties in the case of unbalanced class sizes. For example, let us have two classes with 0.9N data points in the first and 0.1N data points in the second. In this case, we will have $0.18N^{2}$ pairs of points of different classes, 0.9N(0.9N−1) pairs of points in the first class, and 0.1N(0.1N−1) pairs of points of the second class. As a result in (Equation 4) with weights (Equation 5), the attraction of projections of data points from class 1 will dominate the objective function, and the effective projection repulsion and attraction of projections from class 2 will play a negligible role. Changing the α value can not fix the unbalance in the relative influence of attraction in two different classes.

Therefore, it appears reasonable to normalize the weights taking into account the class sizes:(6)$$W_{ij}=\left\{\begin{array}{cc} \frac{1}{2N_{p}N_{r}} & \mathrm{if}L_{p}=l_{i}\neq l_{j}=L_{r}\\ \frac{-\alpha }{N_{r}\left(N_{r}-1\right)} & \mathrm{if}l_{i}=l_{j}=L_{r} \end{array}\right.,$$
where $N_{r}$ is the number of data points of the class with label $L_{r}$. Weight matrix (Equation 6) equilibrates the strengths of projection attraction within each class and the repulsion of projections between two different classes.

More generally, attraction and repulsion between data point projections can be fine-tuned using a priori knowledge about the expected similarity between class labels. For example, this can be the case of ordinal class labels (where there exists a meaningful ranking of class labels). Let us consider the most general form of coefficients of attraction in one class and repulsion in different classes:(7)$$\Delta =\left[\begin{array}{cccc} \delta_{11} & \delta_{12} & \ldots & \delta_{1n}\\ \delta_{21} & \delta_{22} & \ldots & \delta_{2n}\\ \vdots & \vdots & \ddots & \vdots \\ \delta_{n1} & \delta_{n2} & \ldots & \delta_{nn} \end{array}\right]$$

This matrix allows us to define the weight matrix in the following form:(8)$$W_{ij}=\left\{\begin{array}{cc} \frac{\delta_{pr}}{2N_{p}N_{r}} & \mathrm{if}L_{p}=l_{i}\neq l_{j}=L_{r}\\ \frac{\delta_{rr}}{N_{r}\left(N_{r}-1\right)} & \mathrm{if}l_{i}=l_{j}=L_{r} \end{array}\right..$$

The details of a memory-efficient computational implementation of Supervised PCA, as well as the estimation of its computational and memory complexity, are provided in the Appendix A.

It was demonstrated that supervised principal components could substitute several layers of feature extraction deep learning network [33].

### 2.3. Domain Adaptation (DA) Problem for Classification Task

We consider that we have two datasets *X* and *Y*, characterized by the same set of *m* features $\chi_{1},\cdots ,\chi_{m}$. *X* represents a sample from a multivariate distribution *S* that we will call “source domain”, and *Y* is a sample from *T* which we will call “target domain”. In further, we will also call *X* a source dataset and *Y* a target dataset. The dataset *X* is equipped with categorical or ordinal labels attributed to some data points (not necessarily to all of them) from a discrete and not very large set of unique labels.

Essentially, *S* and *T* are characterized by different distributions: *T* is considered to be a transformed or distorted version of *S*. The set of such transformations does not have to be fixed, but it is reasonable to assume some characteristic ones such as

(a) Different from *S* number of samples of each class (different class balance);

(b) Arbitrary shifts and rotations, and changes of scale, representing systemic measurement bias in target domain *T* compared to *S*;

(c) Adding random (in some reasonable definition) noise to some parts of *S*, leading to the displacement of these parts towards the center of the data point cloud, with various degrees of such displacement for different data points;

(d) Different data matrix sparsity patterns between *S* and *T*.

We want to define a sufficiently large number of functions $f_{k}\left(\chi_{1},\cdots ,\chi_{m}\right),k=1,\cdots ,q$ of the variables of the data space where both *X* and *Y* exist, which are optimal in the following sense. Let us select a sufficiently rich family of classifiers (of any convenient type) based on the features $\left\{f_{k},k=1,\cdots ,q\right\}$. We want to optimize the choice of $f_{k}\left(\right)$ functions in order to achieve the maximum performance of the best classifier $C_{1}$ from the family with respect to distinguishing labels in *X* and, at the same time, minimize the performance of the best classifier $C_{opt}$ from the family to distinguish points from *X* and *Y*. In the simplest case, this means that the vectors $f_{k}\left(X\right)$ and $f_{k}\left(Y\right)$ should be similar in some reasonable metrics for every *k*, but, strictly speaking, this does not have to be the general case.

In this study, we are interested in finding a set of optimal for the domain adaptation task *linear* features $\left\{f_{k}\right\}$. At the same time, we do not assume that the family of classifiers should be restricted to linear ones. Indeed, one of the most important applications of linear domain adaptation is to define a restricted set of features $\left\{f_{k},k=1,\cdots ,q\right\}$ which can be used for training a non-linear classifier, obtained as a weighted sum of the initial data variables $\chi_{1},\cdots ,\chi_{m}$ (see examples below).

As usual, from the general considerations, we expect that a set of optimal, with respect to the domain adaptation problem, linear functions $f_{k}$ should sufficiently well approximate the initial dataset *X*. This means that a reasonable approach to finding the optimal functions $f_{k}$ should be based on some kind of adaptation of the PCA problem.

### 2.4. Validating Domain Adaptation Algorithms

According to [1] there are two main ways to validate the results of the application of a domain adaptation algorithm.

We will call direct validation the way which assumes partial knowledge of labels $L_{Y}$ for the “unlabeled” dataset *Y*. In this case, the domain adaptation method is applied to the source set *X* with labels *L* and labels ${\hat{L}}_{Y}$ are predicted for the source dataset *Y*. Afterward, we can use any classification quality measure to evaluate the quality of domain adaptation. Since we can have strongly unbalanced class sizes, we can use balanced accuracy for this purpose:(9)$$BA\left(L_{Y},{\hat{L}}_{Y}\right)=\frac{1}{n}\sum_{r=1}^{n}\frac{\sum_{l\left(y_{i}\right)=\hat{l}\left(y_{i}\right)=L_{r}}1}{\sum_{l\left(y_{i}\right)=L_{r}}1}.$$

The reverse validation idea is different. It does not require any knowledge of “true” labels in *Y* but assumes self-consistency of the solution in the following sense. Firstly we split the source dataset into two parts: the training part $X_{L}$ with labels $L_{L}$ and the test part $X_{T}$ with labels $L_{T}$. then we solve the domain adaptation problem using the dataset $X_{L}$ with labels $L_{L}$ as a source dataset and *Y* as a target dataset. Set of labels ${\hat{L}}_{Y}$ is predicted for all data points in *Y*.

After this step, a reverse problem of domain adaptation is solved, using *Y* as a new source dataset with predicted labels ${\hat{L}}_{Y}$ and $X_{T}$ as a new target dataset. For $X_{T}$, the domain adaptation leads to predicting new labels ${\hat{L}}_{T}$. Afterward, the final step is the calculation of the balanced accuracy $BA(L_{T},{\hat{L}}_{T})$ as in the first case. The obtained accuracy value can also be called “self-consistency” of domain adaptation. One can expect that the value of domain adaptation accuracy as a function of the algorithm hyperparameters depends monotonically on self-consistency. Under this assumption, self-consistency can be used to fine-tune the domain adaptation model parameters. We show the approximately monotonous dependence of accuracy on self-consistency below in some toy examples, but in practice, there is no theoretical guarantee for the universality of such behavior.

## 3. Methods

### 3.1. Semi-Supervised PCA for a Joint Data Set

The main result of this study is introducing a novel linear algorithm of domain adaptation, representing a generalization of Supervised PCA to the case when one has a labeled source dataset *X* and an unlabeled target dataset *Y*. As described above, we look for a common linear representation of *X* and *Y*, in which their multivariate distributions would be as similar as possible, while the accuracy of the classification task (using an appropriate—and not necessarily linear—classifier) for *X* in this representation remains acceptable.

We have a labeled source dataset $X=\{x_{i}\}$ with $N_{X}$ data points, set of labels $l_{i}$ for each point in *X*, and a unlabeled target dataset $Y=\{y_{i}\}$ with $N_{Y}$ points. Let *n* be the total number of unique labels $L_{1},\ldots ,L_{n}$. For uniformity, we define variable z∈X∪Y. Let us define such weight matrix *W* in (Equation 3) that it would lead to achieving the domain adaptation. For this purpose, we would like to introduce effective attraction between projections of *X* and *Y* onto the computed components.

One of the ways to do it is to use the formula for semi-supervised (Equation 4) for *z* with the following weight matrix:(10)$$W_{ij}=\left\{\begin{array}{cc} \frac{\delta_{pr}}{2N_{p}N_{r}} & \mathrm{if}z_{i}\in X,z_{j}\in X,L_{p}=l_{i}\neq l_{j}=L_{r}\\ \frac{\delta_{rr}}{N_{r}\left(N_{r}-1\right)} & \mathrm{if}z_{i}\in X,z_{j}\in X,l_{i}=l_{j}=L_{r},\\ 0 & \mathrm{if}z_{i}\in X,z_{j}\in Y\mathrm{OR}z_{i}\in Y,z_{j}\in X,\\ \frac{\beta }{N_{Y}\left(NY-1\right)} & \mathrm{if}z_{i}\in Y,z_{j}\in Y. \end{array}\right.$$

We can represent this matrix as
(11)$$W=\left[\begin{array}{cc} W^{XX} & W^{XY}\\ W^{YX} & W^{YY} \end{array}\right],$$
where $W^{XX}$ is the matrix of SPCA (Equation 8), $W^{XY}={\left(W^{YX}\right)}^{⊤}$ are zero matrices ($W_{ij}^{XY}=0,\mathrm{for}\;\mathrm{all}\;i,j$), and $W^{YY}=\beta J_{N_{Y}N_{Y}}$, where β>0 is the coefficient of repulsion for the target dataset *Y*.

The modified algorithm is characterized by increased computational time compared to the simple Supervised PCA (see Appendix A). Vector $w^{S}$ (Equation 16) becomes larger, but all additional terms have the same value $\frac{\beta }{NY-1}$ and the required time for this is $T_{\times }$ that is negligible compared to other summands. The first summand in (Equation 4) requires longer summation (additional time is $N_{Y}d^{2}\left(2T_{\times }+T_{+}\right)$). We also need to calculate vector $s_{Y}=\sum_{y\in Y}y$ (additional time is $dN_{Y}T_{+}$). The last additional calculation is the computation of the matrix $Y^{⊤}W^{YY}Y$ and the addition of it to the result (additional time is $d^{2}T_{\times }$ and $d^{2}T_{+}$). Overall, the semi-supervised version of PCA is characterized by the following computational time:(12)$$\begin{array}{cc} t_{SSPCA} & =n\left(nT_{\times }+\left(n-1\right)T_{+}\right)+\left(N_{X}+N_{Y}\right)d^{2}\left(2T_{\times }+T_{+}\right)+dN_{Y}T_{+}\\ \; & +nN_{X}^{2}T_{\times }+d^{2}T_{\times }+\left(n^{2}+1\right)d^{2}T_{+}\\ \; & =t_{m}+N_{Y}d^{2}\left(2T_{\times }+T_{+}\right)+d^{2}T_{\times }+d^{2}T_{+}. \end{array}$$

### 3.2. Supervised Transfer Component Analysis

One of the simplest existing methods of linear domain adaptation is Transfer Component Analysis (TCA) [3], which deals specifically with translation (shifts) of the target domain *T* with respect to the source domain *S*, in a space of features (that can be arbitrary functions of the initial variables).

We can generalize the TCA approach to the case when there exist class labels in the source distribution, similar to the principle of Supervised PCA.

Let us consider augmented datasets $\tilde{X}$ and $\tilde{Y}$ containing the same set of objects as X,Y but characterized by a set of some features produced from the initial datasets X,Y.

Let us denote the means of these datasets as
$${\tilde{\mu }}_{X}=\frac{1}{N_{X}}\sum_{\tilde{x}\in \tilde{X}}\tilde{x};\;{\tilde{\mu }}_{Y}=\frac{1}{N_{Y}}\sum_{\tilde{y}\in \tilde{Y}}\tilde{y}.$$

Now we can write matrix $Q^{W}$ as:(13)$$q_{lm}^{W}=\sum_{i}\left(\sum_{r}W_{ir}\right){\tilde{x}}_{il}{\tilde{x}}_{im}-\sum_{ij}W_{ij}{\tilde{x}}_{il}{\tilde{x}}_{jm}-\phi \left({\tilde{\mu }}_{Xl}-{\tilde{\mu }}_{Yl}\right)\left({\tilde{\mu }}_{Xm}-{\tilde{\mu }}_{Ym}\right),$$
where weights $W_{ir}$ are assigned following the same rules as in semi-supervised PCA (Equation 10), and ϕ>0 is the attraction coefficient between the mean points of the data samples in *X* and *Y*.

For computing the matrix $Q^{W}$ (Equation 13), an accelerated algorithm (Equation 19)–(Equation 26) can be used.

The main advantage of TCA is its low computational complexity. In addition to the semi-supervised PCA (Equation 4), it is necessary to calculate only vectors $s_{X}=\sum_{\tilde{x}\in \tilde{X}}\tilde{x}$ (additional time is $dN_{X}T_{+}$), vectors of means ${\tilde{\mu }}_{X}=s_{X}/N_{X}$ and ${\tilde{\mu }}_{Y}=s_{Y}/N_{Y}$ (additional time $2dT_{\times }$), calculate one more matrix ${\tilde{\mu }}_{Y}^{⊤}{\tilde{\mu }}_{Y}$ (additional time $d^{2}T_{\times }$) and add these matrices to the result (additional time $d^{2}T_{+}$). Therefore, the computational time required for TCA is
(14)$$t_{TCA}=t_{SSPCA}+dN_{X}T_{+}+2d^{2}T_{\times }+d^{2}T_{+}.$$

As we can see from (Equation 12) and (Equation 14), all terms specific for TCA are very small in comparison to $t_{SSPCA}$ (Equation 12).

The choice of features for computing TCA requires special consideration. If the set of features coincides with initial variables $\tilde{X}=X,\tilde{Y}=Y$, then in the resulting projection, the means of the data point clouds will become close. Of course, two data point clouds can be very different even if their mean vectors coincide. Intuitively and under some assumptions, the richer the set of features used to represent X,Y, the more the distributions of their TCA projections will be similar. In the original TCA formulation [3], the features are produced implicitly, using the kernel trick, which is equivalent to minimizing the Minimal Mean Discrepancy (MMD) measure [34] between the TCA projections of *X* and *Y*. This approach leads to an elegant and compact implementation with kernel function hyperparameter. However, the algorithm of TCA can be applied to an arbitrarily produced set of features, even without referring to the kernel-based approach.

### 3.3. Domain Adaptation Principal Component Analysis Algorithm

In applying semi-supervised PCA to the joint dataset X∪Y, there are no effective interactions (neither repulsion nor attraction) between the projections of data points from different domains. In order to reinforce the domain adaptation, we need to make one step further and make the projections of the data points from the target domain to be effectively attracted to similar data points from the source domain. The most non-trivial task here is defining which points from the source domain are similar to a data point from the target domain. In the simplest scenario, we will define such matching through *k*-Nearest Neighbors (kNN) approach. Therefore, we will introduce a term describing the effective attraction of a projection of a data point from *Y* to the *k* closest points from *X* (see Figure 2):(15)$$W_{ij}=\left\{\begin{array}{cc} \frac{\delta_{pr}}{2N_{p}N_{r}} & \mathrm{if}z_{i}\in X,z_{j}\in X,L_{p}=l_{i}\neq l_{j}=L_{r}\\ \frac{\delta_{rr}}{N_{r}\left(N_{r}-1\right)} & \mathrm{if}z_{i}\in X,z_{j}\in X,l_{i}=l_{j}=L_{r},\\ \frac{\beta }{N_{Y}\left(NY-1\right)} & \mathrm{if}z_{i}\in Y,z_{j}\in Y,\\ 0 & \mathrm{if}z_{i}\in Y,z_{j}\notin kNN\left(z_{i}\right)\mathrm{AND}z_{j}\in Y,z_{i}\notin kNN\left(z_{j}\right),\\ \frac{\gamma }{kN_{Y}} & \mathrm{if}z_{i}\in Y,z_{j}\in kNN\left(z_{i}\right)\mathrm{OR}z_{j}\in Y,z_{i}\in kNN\left(z_{j}\right), \end{array}\right.$$
where *k* is the number of the nearest neighbours, kNN(y) is set of *k* labeled nearest neighbours of a data point y∈Y, and γ is the effective attraction coefficient between the projection of y∈Y and the projection of each data point x∈kNN(y).

However, the matching between the data points in two domains using the kNN approach can be strongly affected by the differences between *X* and *Y*, including the simplest translations. Here we deal with a sort of “chicken or egg” problem. To define the neighbors between a data point in *Y* and the data points in *X*, one has to know the best representation of both datasets, so they would be as similar as possible. On the other hand, to find this representation using DAPCA, we need to know the “true” data point neighbors.

As usual, this problem can be approached by iterations. We will use the definition of the nearest neighbors in the initial data variables as the first iteration (alternatively, one can use any other suitable metrics, such as reduced PCA-based representation). It gives us the *q*-dimensional plane of principal components (the eigenvectors of $Q^{W}$) with the orthogonal projector on it $P_{1}$. Afterward, we find for each target sample y∈Y the *k* nearest neighbors kNN(y) from the source samples x∈X in the projection on this plane.

These definitions of the neighbors leads to a new $W_{ij}$, which we use to find the new projector $P_{2}$ and define the new nearest neighbors. Afterward, we iterate. The iterations are guaranteed to converge in a finite number of steps because the functional $H_{W}$ (Equation 4) increases at each step (similarly to *k*-means and other splitting-based algorithms). In practice, it is convenient to use an early stopping criterion that can already produce a useful feature set. Our experiments show that the typical number of iterations can be below 10.

Since the DAPCA algorithm is iterative, estimating its computational complexity is difficult. Of note, the accelerated algorithm’s usage for calculating matrix *W* allows us to calculate only once the constant part of the matrix $Q^{W}$ that corresponds to the semi-supervised PCA and then calculate only the part of $Q^{W}$ related to $W_{XY}$, see Appendix.

### 3.4. Implementation and Code Availability

DAPCA is freely available from https://github.com/mirkes/DAPCA, accessed on 20 December 2022. Both MATLAB and Python implementations are available. The presented implementation allows one to work with large datasets by exploiting several PCA models.

DAPCA model is calculated if a nonempty set of target domain is specified.If the set of points of the target domain is specified as empty set, then the function calculates the Supervised PCA model (see Section 2.2).If the pair `‘TCA’, γ is specified then Supervised TCA with attraction coefficient γ is calculated using the explicitly provided feature space (see Section 3.2).For DAPCA and Supervised PCA models, the repulsion between different classes δ can be specified as:a scalar to have the same repulsion for all classes;a vector *R* with the number of elements corresponding to the number of classes to define the repulsion force between classes *i* and *j* as $\delta_{ij}=\left|R_{i}-R_{j}\right|$;a square matrix with the number of rows and columns corresponding to the number of classes to specify a distinct repulsion force for each pair of classes.

The earlier version of Supervised PCA is freely available from https://github.com/Mirkes/SupervisedPCA, accessed on 20 December 2022. There exists only the Matlab implementation of this function. This implementation is based on constructing the complete Laplacian matrix and, as a result, cannot work with large datasets. This function allows the user to calculate

Standard PCA. This option is not recommended because of the large computation time;Normalized PCA accordingly to paper [26];Supervised PCA accordingly to paper [26] (with α=0);Supervised PCA described in Section 2.2 but with the same repulsion strength for all pairs of classes.

Supervised PCA for regression is freely available from https://github.com/Mirkes/SupervisedPCA/tree/master/Universal%20SPCA%20from%20Shibo%20Lei, accessed on 20 December 2022.

## 4. Results

### 4.1. Neural Architectures Used to Validate DAPCA on Digit Image Data

We used ready Pytorch implementations of the neural network-based classifiers from [1] downloaded from https://github.com/vcoyette/DANN, accessed on 23 September 2021.

### 4.2. Testing DAPCA on a Toy 3-Cluster Example

We synthesized a simple 3D dataset containing two classes of data points representing well-separated unbalanced (by factor 2) clusters in the data point cloud. Each cluster represents a sample from the 3D normal distribution. The parameters of variance were chosen in such a way that both clusters were characterized by a common axis of the dominant variance. A sample from similar but distorted distribution was used as a target domain, with hidden initial class labels (Figure 3A). The distortion included a shift along one of the coordinates such that the degree of shift was different for the two classes. The class balance in the target domain was changed by a factor of 5 such that the number of target domain data points in one class was five times smaller than in the source domain. In addition, we scaled the parameters of variance in each class of data points, scaling by a factor of 2 the variance in one of the classes.

We applied three flavors of PCA described above: PCA, Supervised PCA (SPCA), Domain Adaptation PCA (DAPCA). For each flavor, we computed two first principal components (out of three possible). Neither the standard PCA nor SPCA aligned the source and the target domains, as expected. SPCA produced better-separated classes in the source domain. DAPCA applied with parameters α=1,γ=100 produced a representation of the data in which both source and target domains were well aligned and at the same time the class labels in the source domain were well separated (see Figure 3B).

DAPCA results were stable in a large interval of the parameters α,γ (Figure 3C). We also found that the number of nearest neighbors in the kNN graph is not a sensitive parameter. The number of iterations of the DAPCA algorithm producing the correct alignment of the source and target datasets was approximately ten.

We used the simplest support vector classifier (SVC) in order to predict labels in the target domain using known labels in the source domain. The classifier was trained using the linear features computed by DAPCA. Since in the toy example we knew the hidden labels in the target domain by design, we could estimate both accuracy and self-consistency measures of domain adaptation as described in the Methods section (Figure 3C). The pattern of computed self-consistency in a range of parameter values α,γ was informative for anticipating the balanced accuracy of the prediction (correlation coefficient around 0.75). The combination of parameters leading to the large self-consistency value corresponded to the large prediction accuracy. However, the opposite was not true, small values of self-consistency can correspond to both high and small accuracy.

Using the same toy example, we compared several linear domain adaptation methods, listed in Table 1. The toy example is designed in such a way that the projections on the first principal component do not separate well the classes neither in the source nor in the target domain. In addition, the covariance matrices and the class balance are not exactly the same in the source and the target. As a result, those linear methods of domain adaptation that do not take into account class labeling information struggled to align the 2D projection distributions, and DAPCA was the only method that resulted in a good alignment of two classes, see Figure 4. The Python notebook with the code of this test is provided at https://github.com/mirkes/DAPCA.

### 4.3. Validation Test Using Amazon Review Dataset

We made a standard for domain adaptation field validation of the DAPCA method using the same Amazon review dataset as was used in [1], see Figure 5. The dataset represents a set of items of various kinds (books, kitchen, dvd, electronics), characterized by text reviews and the annotated binary sentiment score (positive or negative). The text of the review is represented as a vector in a multi-dimensional space by using an embedding method that produces numerical features that can be ordered by their information importance. In our experiments, we took the first 1000 features from a small Amazon reviews subset obtained from https://github.com/GRAAL-Research/domain_adversarial_neural_network, accessed on 10 December 2021. We trained a simple logistic regression either on the full set of 1000 features or using the reduced dataset with PCA, SPCA or DAPCA. The regression was trained using the labels for one item type as a source domain and then tested using the items of another type as a target domain. In most pairwise comparisons between item types, DAPCA provided the best set of features for the classification of items in the target domain, see Figure 5.

### 4.4. Validation Using Handwritten Digit Images Data

Domain Adaptation Neural Network (DANN) approach was shown to obtain impressive results for solving the domain adaptation problem for the task of digit image classification [1]. The DANN architecture (see Figure 6A) combines the convolutional neural network (CNN) with an adversarial classifier in order to extract such a representation of the digit images that would allow achieving good classification in the source domain (for example, in the MNIST dataset) and as close as possible multivariate distributions in the source and target domains (for example, the SVHN dataset of images taken from real street photos together with their backgrounds).

We compared the performance of the DANN neural network architecture with a much lighter one, where DAPCA was used for domain adaptation of the common representation of both source and target domains but which was learned by a standard CNN part of DANN using only the source domain and its labels. DAPCA was applied in the space of the features provided by the very last layer of the classifier which could be used to predict the image label using simple logistic regression.

In our experiments, we used two tasks with two pairs of source and target domains (MNIST vs. MNIST-M and SVHN vs. MNIST). We used the components computed with DAPCA on the features learned using the source domain only and recorded from the last layer of the CNN part of the DANN architecture. For applying the logistic regression to predict the image labels, we used as many first DAPCA components as corresponded to non-negative eigenvalues of the spectral decomposition. We visualized the multidimensional data point cloud containing the points from the source and the target domains using Uniform Manifold Approximation and Projection for Dimension Reduction (UMAP) method. Namely, we compared three visualizations of the digit image representations: (1) one obtained by training the CNN part of the neural network and using the source domain only; (2) one obtained by applying DAPCA on top of (1), using the target domain without labels; (3) one obtained through application of the full DANN architecture, using both source domain with labels and target domain without labels. The results are shown in Figure 6C,D.

We concluded that the performance of domain adaptation provided by DAPCA is relatively modest in this benchmark, especially, when compared to the full non-linear DANN architecture. The advantage of using the domain adaptation is usually measured by how much the classifier approaches the theoretically maximum accuracy $A_{top}$ on the target domain which is measured in benchmarks by training the classifier directly on the known labels in the target dataset. If one indicates as $A_{noDA}$ the accuracy of the classifier trained on the source domain without any domain adaptation (standard CNN in this case) then the benefit of using domain adaptation can be measured as $b=\frac{A_{DA}-A_{noDA}}{A_{top}-A_{noDA}}$, where $A_{DA}$ is the performance of the classifier with domain adaptation. The meaning of *b* is the fraction of the gap between the performance of a classifier without domain adaptation and the theoretical maximal performance achievable if all labels in the target domain would be known. Thus, in the task with MNIST dataset as the source and MNIST-M as the target domains, DAPCA resulted in b=8% compared to b=79% obtained by the DANN architecture. In another example (SVHN as the source and MNIST as the target domains), DAPCA resulted in b=23% while DANN resulted in b=57%. Such modest performance of DAPCA compared to DANN can be explained by that the most of the learning in the DANN architecture from Figure 6A is happening in the convolutional layers of the feature extractor part and this learning uses examples from the target domain. DAPCA-based domain adaptation shown in Figure 6B does not use at all the examples from the target domain for learning the image representation, so the result is not surprising. On the other hand, we can document a measurable and significant benefit from applying DAPCA in the domain adaptation tasks at the very late layers of the neural network. This means that potentially a variant of DAPCA can be used as a layer on top of a convolutional feature extractor which can be trained (similarly to the famous Deep CORAL approach [15]), but building such an architecture bypasses the focus of the current study.

Of note, training the DANN architecture shown in Figure 6A is rather heavy (tens of hours on CPU), while the DAPCA-based domain adaptation shown in Figure 6B requires much less time (few minutes on CPU).

We repeated the same benchmark in the context of a small sample size by using subsampled digit image datasets (we used only 3000 images for training and testing in each domain). The qualitative conclusions remained unchanged: the simple DAPCA-based solution was less performant than the full-scale DANN architecture even if the difference between the corresponding performances was less striking.

### 4.5. Application of DAPCA in Single-Cell Omics Data Analysis

To demonstrate that our approach is not limited to typical machine learning applications, we also applied DAPCA in the context of single-cell RNA-seq (scRNA-seq) data analysis. scRNA-seq data is obtained by measuring the abundance of messenger RNA (mRNA) transcripts expressed within the individual cells contained in a sample (e.g., biological tissue or an in vitro cell culture). It yields for each individual cell a molecular profile represented as a long integer-valued vector, which contains for each gene the number of transcripts measured. In this framework, a sample can therefore be seen as a data point cloud of its individual cells. Single-cell data science is an active research domain [35] where the application of dimensionality reduction techniques is of utmost importance, as the data spaces typically possess very high dimensionality (tens of thousands of features). In particular, PCA is widely used to reduce the initial tens of thousands of variables to only a few tens of principal components, and most of the analyses (clustering, classification, or more generally all metric-based methods) are performed within this reduced space to mitigate the limitations linked to the curse of dimensionality.

One of the most important challenges in single-cell data analysis is the presence of dataset-specific biases, caused by a variety of factors: experimenter differences, variations in the genetic and environmental background when dealing with data coming from different individuals, sequencing technologies, etc. The consequence of these biases is that data point clouds associated with different datasets appear to be displaced with respect to one another, and require special alignment procedures often referred to as *horizontal data integration* or *batch correction* [36]. In this application, we will demonstrate the capabilities of DAPCA to serve as a batch correction algorithm for scRNA-seq data.

We used three annotated, public single-cell datasets of normal lung tissue [37]. Lung tissues were obtained from patients undergoing lobectomy for local lung tumors. According to [37], patient 1 was a 75-year-old male diagnosed with an early stage adenosarcoma; patient 2 was a 46-year-old male diagnosed with an endobronchial carcinoid tumors; patient 3 was a 51-year-old female with mild asthma diagnosed with an endobronchial carcinoid tumors. Epithelial and immune cells were first filtered using magnetic-activated cell sorting (MACS) and sorted using a cell sorter into four categories (epithelial, immune, endothelial or stromal). Sorted cell libraries were prepared using the 10X Genomics 3’ Single Cell V2 protocol, then pooled and sequenced on a Novaseq 6000 (Illumina). According to the authors, reads were demultiplexed using Cell Ranger v2.0, and cells with fewer than 500 genes or 1000 UMIs were discarded, ending up with 65,667 valid cells.

We preprocessed the three raw count matrices independently. First, each cell was normalized to 10,000 counts, and log(1+x) transformation has been applied according to the existing preprocessing standards. We pooled each cell to the average of its 5 nearest neighbors (using Euclidean metric in the space of the dataset’s 30 first principal components) to reduce noise. Eventually, we selected the 10,000 most variable genes in each dataset to end up with three preprocessed expression matrices, each expressed in a 10,000 feature space.

Lung tissue contains a complex, hierarchical population of cells of various types and states organized into different compartments. Strong differences in specific genes expressed in each of these compartments cause cell-associated gene expression vectors to form more or less compact clusters in the multi-dimensional gene space (see Figure 7A,B. We can see data point clouds corresponding to the three lung datasets do not overlap even when looking at cells from different datasets associated with the same compartment. We suggest using DAPCA instead of standard PCA to carry out dimensionality reduction, taking into account the author-provided cell annotations (endothelial, stromal, epithelial and immune). We set the first dataset to be the source domain, as it contains the largest number of cells, and we consider the union of the two other datasets to be the target domain. DAPCA is aimed at finding a low-dimensional linear projection (with only a few tens of features) in which the multivariate distribution of projections from different samples would appear as similar as possible, while cells with labels related to different compartments would be maximally separated.

Application of DAPCA in such context is shown in Figure 7A. Visual inspection of the resulting projections into the first 30 components extracted by PCA, SPCA and DAPCA and further visualization using UMAP shows that the DAPCA projections overlap much better than PCA and SPCA projections (Figure 7A, top panels). At the same time, the separation between cell types remains well preserved (Figure 7A, bottom panels).

In order to quantify the effect of domain adaptation, we trained a simple kNN classifier (k=20) to predict the dataset of origin of each cell within the DAPCA representation. We expect the classifier to perform poorly when domain adaptation is successful, meaning that the source and target datasets are indistinguishable. It also makes sense to normalize the performance of such classifier with respect to its baseline level accuracy which can be estimated by randomly permuting the labels of the datasets. Both absolute and normalized accuracies are shown in Figure 7C. Comparison between PCA, SPCA and DAPCA using this strategy allows confirming that DAPCA outperforms the two other methods at making cells less distinguishable with respect to their dataset of origin.

We also observe DAPCA does not merge equally well the cells belonging to different compartments (Figure 7C). For instance, domain adaptation applied to endothelial cells appears to be close to the theoretical optimal performance. On the other hand, domain adaptation applied to the cells from the stromal compartment was less good. This could be explained by the high heterogeneity within the cells annotated as stromal, which are grouped into four different clusters. We followed this first analysis step by extracting the subparts of the datasets corresponding to the stromal cells and we applied PCA, SPCA, and DAPCA to this subset of cells. In order to apply SPCA and DAPCA we defined new labels in Dataset 1 serving the source domain, by clustering it with the standard Louvain clustering algorithm (these clusters are shown in color in Figure 7B) such that the clusters hypothetically correspond to major subpopulations within the stromal cell compartment. Four such subpopulations have been identified. The DAPCA-based domain adaptation in this case shows the performance close to be optimal (Figure 7D). Interestingly, three out of four clusters seemed to match and at least partially mix with the cells from the target domain (Datasets 2 and 3). One of the clusters appeared to remain specific to the source domain (Dataset 1), and could correspond to a subpopulation of lung cells specific to dataset 1, and located in the stromal compartment.

Overall, we can conclude that DAPCA can be used as a tool for simultaneously integrating scRNA-seq datasets from different origins as well as reducing their dimensionality, as long as cell annotations are available for at least one dataset. We furthermore showed DAPCA is able to preserve cell–cell similarity in a biological sense, meaning cells within similar compartments and expression profiles remain close to one another after the algorithm application. Compared to other widely used techniques, DAPCA is based on linear dimensionality reduction which does not tend to overfit the data integration task. In particular, it naturally allows one to consider the existence of specific parts in the source or in target domains that can have specific biological properties and should not be easily matched between the source and the target domains. In addition, we show that DAPCA transformation of the data can be computed locally with respect to a subpart of the data point cloud which might lead to better performance than the global domain adaptation.

## 5. Discussion

In this paper, we suggest a novel base linear method for solving the problem of domain adaptation which can serve as a preprocessing step for the application of more sophisticated and non-linear machine learning approaches. The method is named Domain Adaptation Principal Component Analysis (DAPCA) because it represents principal component analysis generalization. As input DAPCA takes a pair of two datasets (source *X* and target *Y*), one of which is labeled (source) and another is not (target). Formally, one of these datasets can be empty. If the target domain *Y* is empty then DAPCA degenerates to the supervised PCA in the source domain *X*. If the source domain *X* is empty, DAPCA degenerates to the standard PCA in the target domain. If the source domain *X* contains only one label then DAPCA represents a specific version of consensus PCA which can be used to solve the data integration task. The classical domain adaptation problem (which is sometimes called unsupervised in the sense that no label information is available for *Y*) can be extended to the semi-supervised case (where partial information on labels in *Y* is known). DAPCA can be easily adapted to this situation, too, by introducing the proper weighting schema between pairs of data points.

A large number of variables characterizes many modern datasets so that the corresponding data point clouds formally exist in high- or very high-dimensional space. A typical step in analyzing such datasets is a dimensionality reduction step to a more manageable number of dimensions (e.g., few tens or even 3–4). For example, this is the typical case of omics data in the field of biology, including single-cell data [38,39]. If this number is close to an estimated intrinsic dimensionality [40,41] of the data, then this step does not lead to a significant loss of information. The reduction is frequently made through the use of the classical PCA. DAPCA allows a user to easily replace this step when the divergence between the source and the target datasets is suspected. In addition, it takes into account the labeling data. The iterative DAPCA also helps to resolve the classical distance concentration difficulty (curse of dimensionality): in really large dimensional distributions, the kNN search may be affected by the distance concentration phenomena: most of the distances are close to the median value [42]. It was shown that the use of fractional norms or quasinorms does not save the situation [43]. However, dimensionality reduction may help to overcome this.

DAPCA is based on a data point matching step, where for each point from the target dataset one has to indicate the most similar, with appropriate metrics, data points from the source dataset. In the current implementation, the simplest kNN approach is used for this purpose, but this step can be more sophisticated. Some ideas can be borrowed from known methods of data fusion in machine learning. It can be the use of mutual (reciprocal) nearest neighbors, or the application of optimal transport-based algorithms for matching the points in two finite data point clouds [18].

Supervised PCA and DAPCA can also be used as fast preprocessing steps for unsupervised non-linear methods of data analysis and data approximation, enabling them to take into account the data point labeling information. Therefore, they can make other methods at least partially supervised. For example, elastic principal graphs [44,45], self-organizing maps [46], UMAP [47], t-SNE [48], or Independent Component Analysis [29] can directly benefit from DAPCA or SPCA as preprocessing steps. Such an approach can find applications in many domains, such as bioinformatics or single-cell data science [35].

As it was expected, our study shows that neural-network-based classifiers equipped with an adversarial module that tries to distinguish the source from the target domains (such as DANN) achieve better performance than the linear and more constrained DAPCA approach when tested on imaging data. This is partly explained by the fact that the convolutional layers of DANN are trained on the information from both source and target domains, while in our comparison DAPCA used the image representation trained on the source domain only. Linear methods such as DAPCA are deterministic, computationally efficient, reproducible, and relatively easily explainable. Therefore, the linear approaches occupy a niche in those machine learning applications where such features are more important than the maximum possible accuracy. Training neural networks and especially choosing their architectures remains an art, requiring intuition, experience, and a lot of computational resources, but this can lead to superior results, in terms of accuracy. In a sense, DAPCA stays in the same position in DANN as PCA to the auto-associative neural networks (neural-network-based autoencoders) [49]. However, PCA was introduced almost a century before the neural-network-based autoencoders, while a standard fully deterministic and computationally efficient linear approach to domain adaptation based on optimization and using labels from the source domain, is still lacking. Introducing Domain adaptation PCA fills this gap.

DAPCA, like any other method of domain adaptation, has certain limitations in some data analysis scenarios. The application of DAPCA requires the user to specify the values of several hyperparameters (the strength of the attraction force between the points of the same class, the attraction force between the domains, and the number of the nearest neighbors as the most important ones). Even though these parameters might take some recommendations from the practice values, it might still be required to do some fine-tuning in a concrete application. Therefore, in simple situations, other and simpler linear approaches for domain adaptation might have similar to DAPCA performance but be more convenient in applications. When an essentially non-linear encoding of the input data is needed (as in the case of the image data analysis), neural-network-based architectures might be a preferable choice on the other side of the regularity-flexibility trade-off. DAPCA is, by design, more difficult to integrate as a component into more complex deep classifiers, compared to some other linear domain adaptation approaches. Enabling this option can be an important direction for future work.

Nevertheless, we have clearly demonstrated that applying DAPCA might be preferable to other methods in certain scenarios. For example, we showed that its application would be beneficial when both source and target domains are characterized by important sources of variance, which do not coincide with the hyperspace where the best separation of classes is achieved. Such a situation is rather typical in analyzing omics data in biology, where the first principal components are frequently associated with the technical or irrelevant sample classification biological factors.

Therefore, we are confident that DAPCA can be a useful method in the toolbox of domain adaptation methods, and definitely there exist niches where we expect the application of DAPCA to be preferred to other existing methods of domain adaptation.

## Figures and Tables

**Figure 1 entropy-25-00033-f001:**
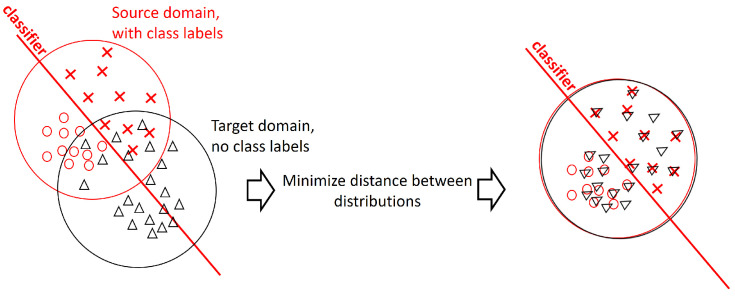
The idea behind domain adaptation learning. The source domain has labels and can be used to construct a classifier. The target domain where the classifier is supposed to work does not have labels. It is suggested to find a common representation of two domains such that their distributions would maximally match each other, and simultaneously build the efficient classifier using this representation and available labels.

**Figure 2 entropy-25-00033-f002:**
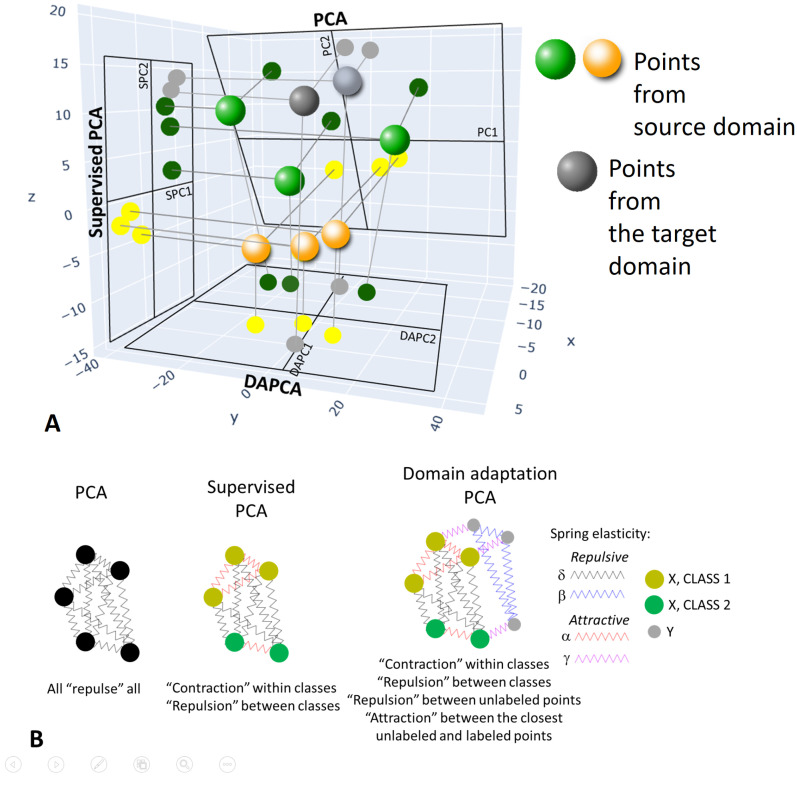
Illustration of the Domain Adaptation PCA (DAPCA) principle. (**A**) PCA, Supervised PCA and DAPCA provide three different ways to reduce the data dimensionality by a linear projection. DAPCA considers both labeled and unlabeled datasets and computes such projection that the projection distributions would be as similar as possible. (**B**) Minimizing the quadratic functional for finding each linear projection can be interpreted as introducing repulsive and attractive forces between data point projections. Of course, data points (shown as 3D spheres) do not repulse or attract, remaining fixed; therefore, the terms ’repulsion’ or ’attraction’ are quoted in this Figure’s text. PCA can be interpreted as a result of effective repulsion between all data point projection pairs. In projection onto the Supervised PCA plane, the scattering within a data point class is minimized while the scattering between the classes is maximized. This can be interpreted as the effective attraction of data point projections for the data points of the same class. In DAPCA, four types of effective “forces” exist between data point projections: repulsive in source and target datasets, attractive between data points of the same class in the source dataset, attractive between the data points in the target and the closest data points in the source dataset.

**Figure 3 entropy-25-00033-f003:**
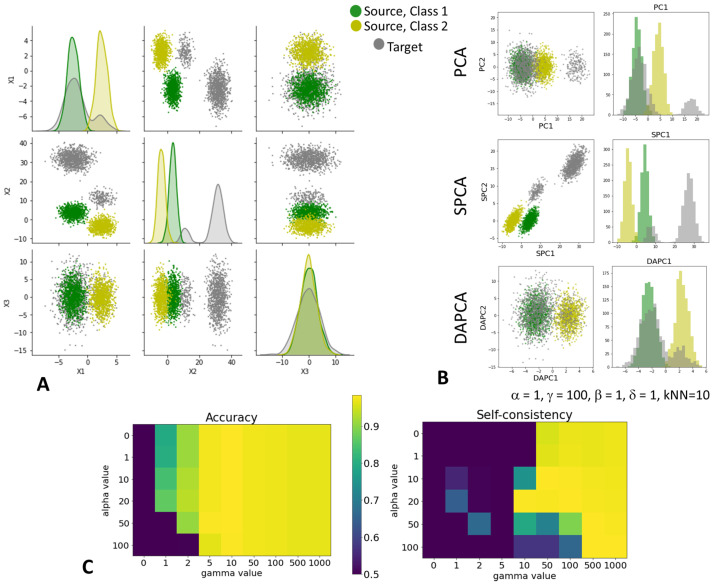
Toy 3D dataset used to test the DAPCA algorithm. (**A**) Configuration of data points of two classes in the source domain (green and yellow data points) and in the target domain (grey data points). The target domain distribution differs from the source domain by a shift along the second coordinate (the degree of the shift is different for two classes of the source domain), by the different balance of class composition and by the different variance scales within each class. (**B**) Application of three flavors of PCA, showing the projections onto the first two principal components (on the left) and the histogram of projections on the first principal component (on the right). (**C**) Comparing the accuracy of predicted labels in the target domain and the self-consistency of domain adaptation by DAPCA for a range of key DAPCA parameters.

**Figure 4 entropy-25-00033-f004:**
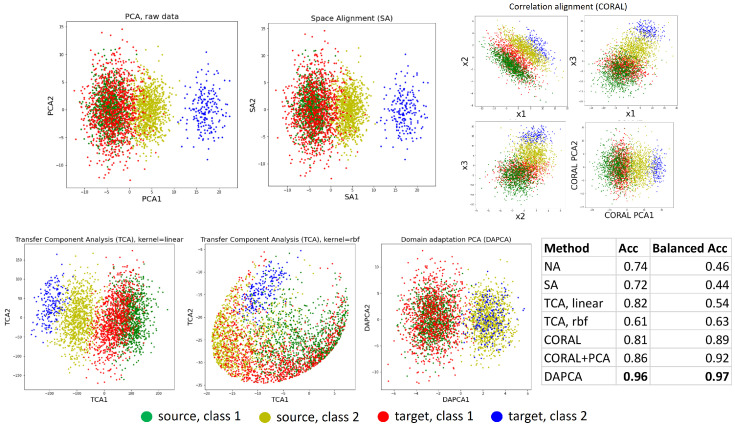
Comparison of linear domain adaptation methods, using the two classes toy example from Figure 3. For CORAL, projections on all three dimensions are shown together with PCA, because CORAL does not reduce the data dimensionality. Therefore, CORAL accuracy was computed in the full feature space (marked as CORAL in the table) and after reducing the dimensionality of the merged source and target datasets transformed by CORAL (marked as CORAL+PCA). The accuracy of the domain adaptations task was estimated with known ground-truth target domain labels, using the standard Support Vector Classifier implementation in sklearn, run with default parameters. The bold font indicates the maximum accuracy.

**Figure 5 entropy-25-00033-f005:**
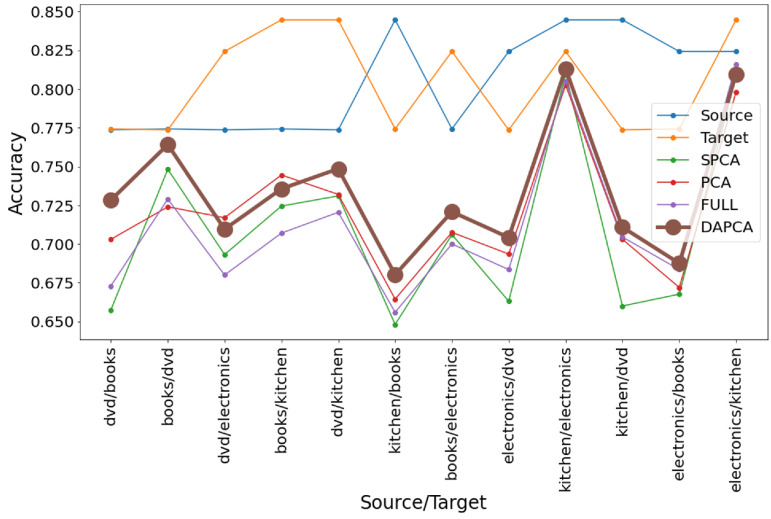
Validating DAPCA using Amazon review dataset. Source and target lines indicate the performance of the prediction separately on the source and target domains (without domain adaptation). Other lines correspond to the performance of logistic regression trained on different features: all features (FULL), PCA, SPCA, and DAPCA (200 top components were taken for each method). DAPCA parameters used here were α=0,γ=1,kNN=5.

**Figure 6 entropy-25-00033-f006:**
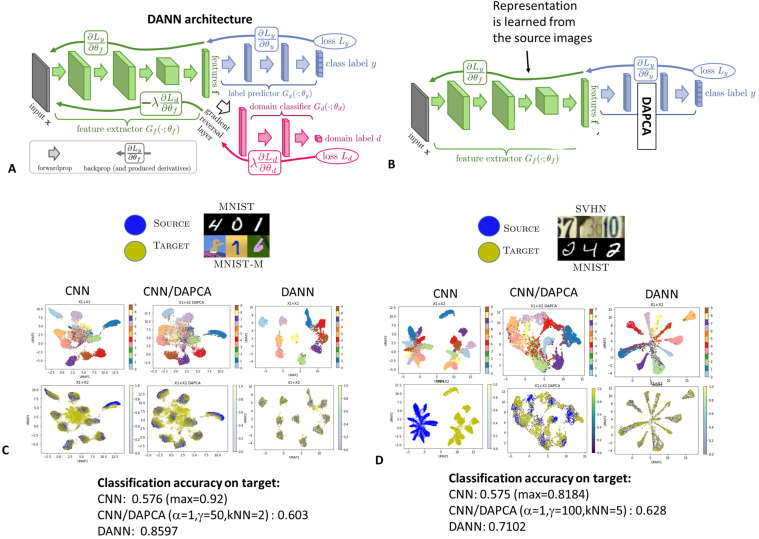
Validation of DAPCA in digit image classification using two distinct domains. (**A**) the original DANN adversarial learning-based architecture for solving the domain adaptation task. The image is adapted with permission from [1]. (**B**) Simplified DAPCA-based architecture for domain adaptation. The domain adaptation is performed for the features recorded from the last layer of the neural network before applying the last classification step, which can be replaced with logistic regression. (**C**,**D**) Computing the domain adaptation benefit for several architectures: CNN: no domain adaptation, CNN/DAPCA: as shown in panel (**B**), DANN: adversarial learning-based domain adaptation. UMAP visualizations of internal image representations from the source and the target domains are shown on the plot. The text reports the accuracy of classification from these representations using logistic regression. “Max” specifies the maximum achievable accuracy when the CNN classifier is trained directly on the target domain with known labels.

**Figure 7 entropy-25-00033-f007:**
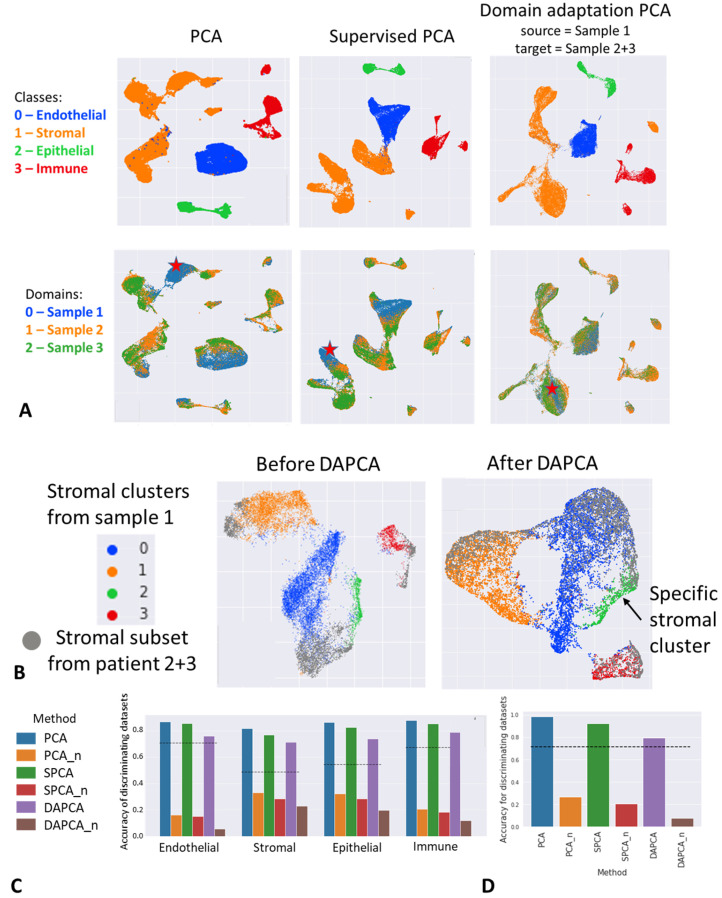
Application of DAPCA for the task of integrating single-cell datasets (three healthy lung tissue samples, in this case, the count data is used with permission from [37] using the publicly available URL https://www.synapse.org/Synapse:syn21041850/files/, accessed on 20 December 2022). (**A**) The result of the global application of DAPCA to all data points in three domains. Top panel: UMAP visualizations on top of 30 components extracted by PCA, SPCA, and DAPCA with colors corresponding to the major cell type annotations. Bottom panel: same as the top panel but with colors corresponding to three different samples. A cluster of data points from Sample 1 is marked by a red star which appears to be dataset-specific in the PCA projection. This cluster becomes well-integrated in the target domain in the DAPCA projection. (**B**) Application of DAPCA locally to a subpart of the cell populations in three samples (only stromal cells). The labels in the source domain are defined here through Louvain clustering of the source domain (blue, orange, green, and red colors). The panel “After DAPCA” shows the UMAP visualization on top of 30 components computed by DAPCA, from which one can determine the existence of a sample-specific cluster of cells (green color) in the source domain (Sample 1) that does not match any other clusters in the target domain. (**C**,**D**) Measuring the performance of domain adaptation tasks for global and local applications of DAPCA correspondingly. Suffix “_n” indicates the normalized performance computed in the way described in the text. The smaller the accuracy of the kNN classifier trying to distinguish between the samples, the better the domain adaptation task was solved. In particular, close to zero normalized performance of the classifier indicates theoretically maximal domain adaptation, as could be achieved by permuting the labels corresponding to samples.

**Table 1 entropy-25-00033-t001:** Summary and comparison of linear Domain Adaptation methods. PCA and SPCA do not solve the domain adaptation task but are listed here for convenience of comparison.

Method Name	Reference	Principle	Optimi-zation-Based	Low Dimensional Embedding	Use Class Labels in Source
Principal Component Analysis (PCA)	[20]	Maximizes the sum of squared distances between projections of data points.	yes	yes	no
Supervised Principal Component Analysis (SPCA)	[21], this paper	Maximizes the difference between the sum of squared distances between projections of data points in different classes, and the sum of squared distances between projections of data points in the same classes (with weights)	yes	yes	yes
Transfer Component Analysis (TCA)	[3]	Minimizes the Minimal Mean Discrepancy measure between projections of source and target	yes	yes	no
Supervised Transfer Component Analysis (STCA)	this paper	The optimization functional is the one of SPCA plus the sum of squared distances between the mean vectors of projections of data features is minimized	yes	yes	yes
Subspace Alignment (SA)	[13]	Rotation of the *k* principal components of the source to the *k* principal components of the target	no	yes	no
Correlation Alignment for Unsupervised Domain Adaptation (CORAL)	[14]	Stretches the source distribution onto the target distribution. The source is whitened and then “recolored” to the covariance of the target, without reducing dimensionality. The transformation of the source optimizes the functional $\min_{A}\left|\right|A^{T}C_{s}A-C_{t}{\left|\right|}_{F}$, where $C_{s},C_{t}$ are the source and target covariance matrices.	yes	no	no
Aggregating Randomized Clustering-Promoting Invariant Projections	[16]	Minimizes the dissimilarity between projection distributions plus the mean squared distance of the data point projections from the centroids of their corresponding classes. Pseudolabels are assigned to the target via randomized projection approach and majority vote rule	yes	yes	yes
Domain Adaptation PCA (DAPCA)	this paper	DAPCA Maximizes the weighted sum of three following functionals: (a) For the source: the sum of squared distances between projections of data points in different classes and the weighted sum of squared distances between projections of data points in the same classes (as in SPCA). (b) For the target: the sum of squared distances between projections of data points (as in PCA). (c) Between source and target: the sum of squared distances between projection of a target point and its *k* closest projections from the source.	yes	yes	yes

## Data Availability

Only publicly available datasets were analyzed in this study. The Amazon reviews dataset was obtained from https://github.com/GRAAL-Research/domain_adversarial_neural_network. The single cell transcriptomic count data was obtained from https://www.synapse.org/Synapse:syn21041850/files/. The digit images data was obtained using instructions from https://github.com/vcoyette/DANN.

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
