# Peer review of "Domain Adaptation Principal Component Analysis: Base Linear Method for Learning with Out-of-Distribution Data"

_entropy, 2022, doi:10.3390/e25010033_

Round 1
Reviewer 1 Report
This paper proposes a dimension-reduction method for domain adaptation tasks that generalizes the supervised principal component analysis (PCA) in the source domain to the unlabeled target domain.
Overall, this paper is not easy to follow, there exist many missing references ([?]) in this manuscript. Besides, the equation in the method section is also not well explained. The authors are strongly suggested to provide a neat explanation of their method (background knowledge should be separated from the proposed method). And the difference with previous methods like TCA is not well described. In addition, other works [1,2] using pseudo labels in the unlabeled target domain always perform better than TCA, which are not even discussed and compared in this paper.
[1]. Long, Mingsheng, et al. "Deep transfer learning with joint adaptation networks." International conference on machine learning. PMLR, 2017.
[2]. Liang, Jian, et al. "Aggregating randomized clustering-promoting invariant projections for domain adaptation." IEEE transactions on pattern analysis and machine intelligence 41.5 (2018): 1027-1042.
Author Response
This paper proposes a dimension-reduction method for domain adaptation tasks that generalizes the supervised principal component analysis (PCA) in the source domain to the unlabeled target domain.
Overall, this paper is not easy to follow, there exist many missing references ([?]) in this manuscript.
Answer: we are sorry for this. After checking it appeared that all missing references were connected to incorrect translations of the BibTeX items representing websites (GitHub repositories). We corrected this and made sure there is not a single missing reference in the final manuscript.
Besides, the equation in the method section is also not well explained. The authors are strongly suggested to provide a neat explanation of their method (background knowledge should be separated from the proposed method).
Answer: We are thankful to the reviewer for this suggestion. We reorganized the structure of the manuscript by introducing a new section ‘Background’ which describes previously known concepts and ideas, and we left in the Methods section the description of the proposed DAPCA algorithm. The Results section now contains only the results of experimenting with DAPCA and comparing it with other algorithms on synthetic and real-life datasets.
And the difference with previous methods like TCA is not well described.
Answer: We agree that it can be difficult for a reader to distinguish between different members of the whole family of linear domain adaptation methods. In order to facilitate the reading, we introduced Table 1, which provides a summary of the principles exploited by each of the most popular linear domain adaptation methods.
In addition, other works [1,2] using pseudo labels in the unlabeled target domain always perform better than TCA, which are not even discussed and compared in this paper.
[1]. Long, Mingsheng, et al. "Deep transfer learning with joint adaptation networks." International conference on machine learning. PMLR, 2017.
[2]. Liang, Jian, et al. "Aggregating randomized clustering-promoting invariant projections for domain adaptation." IEEE transactions on pattern analysis and machine intelligence 41.5 (2018): 1027-1042.
Answer: These are indeed important references in the domain adaptation field, we are sorry that we missed them. Now we cite and discuss these references in the Introduction together with the CORAL approach mentioned by another reviewer.
Reviewer 2 Report
Presented is a linear method for domain adaptation using PCA. Its performance compared to DANN, which is a fair control. The paper is well written, the motivation is clear.
A couple of potential issues: one, there are already very popular linear methods for doing domain adaptation (DA), amongst them is CORAL (Sun, Feng and Saenko), that finds a transformation on the features of the source distribution that minimizes the L2 norm between it and the target distribution. This paper should be both cited and compared to.
The main purported benefit of this method is its speed. Given this, why are results not performed on a standard DA dataset for vision like VisDA? There should be no GPU / compute constraints that preclude this.
Another concern is the hand construction of features. Correctly setting k for the KNN, the attraction force, which principal components to use, etc, all of these could be learned by a model, which is essentially what DANN does.
Author Response
Presented is a linear method for domain adaptation using PCA. Its performance compared to DANN, which is a fair control. The paper is well-written, and the motivation is clear.
A couple of potential issues: one, there are already very popular linear methods for doing domain adaptation (DA), amongst them is CORAL (Sun, Feng and Saenko), that finds a transformation on the features of the source distribution that minimizes the L2 norm between it and the target distribution. This paper should be both cited and compared to.
Answer: This is a good suggestion that we had overlooked mainly because CORAL represents an efficient approach to solving domain adaptation in a fully unsupervised way, without using the labels from the source domain. Our focus was on the case when the labels from the source domain are used. Moreover, DAPCA represents a dimensionality reduction approach while CORAL de facto not.
Now we discuss CORAL in the Introduction and also introduce Table 1, explaining the most important differences between the most popular existing linear domain adaptation methods. In addition, we compared CORAL with TCA, SA and DAPCA using the toy example that we previously suggested in the manuscript, and deposited the comparison code in the DAPCA manuscript. All this is described in the revised version of the paper.
The main purported benefit of this method is its speed. Given this, why are results not performed on a standard DA dataset for vision like VisDA? There should be no GPU / compute constraints that preclude this.
Answer: We agree that demonstrating DAPCA in applying to VisDA datasets would be an interesting exercise. However, from our experience, dealing with image data and especially preprocessing and encoding them in order to meaningfully apply machine learning methods requires specific technical expertise that none of the authors necessarily have at a sufficient level. Therefore, proper benchmarking of DAPCA based on imaging data did not look realistic in the suggested timeframe of the manuscript revision. In addition, we have a feeling that different linear domain adaptation methods solve different tasks (eg, unlike DAPCA most of them do not take into account the labels in the source data as summarized in the new Table 1). We limited ourselves by comparing popular linear domain adaptation methods in a synthetic example (new Figure 4).
Another concern is the hand construction of features. Correctly setting k for the KNN, the attraction force, which principal components to use, etc, all of these could be learned by a model, which is essentially what DANN does.
Answer: We agree with this remark, and from what is described in the manuscript, DANN indeed achieves impressive results in some benchmarking examples. However, from the general principles, DANN requires a substantial number of examples to train the adversarial classifier, and it is also computationally expensive. Moreover, the DANN approach does not represent a fixed architecture, its adversarial part must be configured for the best performance in a concrete source/target example. In addition, in real practice, it remains difficult to judge if DANN solves the domain adaptation problem efficiently without knowing the ground truth labels in the target domain. In order to highlight the different niches occupied by linear and neural-network-based methods in machine learning applications, we introduce a paragraph in the Discussion section.
Fine-tuning hyperparameters in DAPCA (essential ones which the Reviewer correctly identified) is indeed required but as is the case for many linear and non-linear methods, a suboptimal choice of them can be recommended from practice in certain application fields such as the analysis of single-cell data (for some other base methods like PCA such recommendation for the parameter values do exist). This recommendation should come from the practice of using DAPCA. The big advantage of DAPCA, however, is that it represents a simpler, determenistic and computationally efficient method compared to DANN, which might outweight the fact that it provides less flexible data approximators compared to very flexible neural networks.
Reviewer 3 Report
The paper needs careful proof reading because of numerous grammatical and spelling errors. Although the paper provides many figures the results are too cryptic. For example, in Figure 6, two classification accuracies are presented: 0.576 and 0.92 (max). Why provide both and what does max refer to? Also, it seems that DANN outperforms the CNN/DAPCA. Is this correct?
Author Response
The paper needs careful proof reading because of numerous grammatical and spelling errors.
Answer: We made an extensive and thorough grammatical and spelling check, all the introduced modifications are marked in a red color font in the revised version of the manuscript.
Although the paper provides many figures the results are too cryptic. For example, in Figure 6, two classification accuracies are presented: 0.576 and 0.92 (max). Why provide both and what does max refer to?
Answer: We are thankful for this remark. "Max" specifies the maximum achievable accuracy when the CNN classifier is trained directly on the target domain with known labels. Now it is clearly indicated in the legend for the figure.
Also, it seems that DANN outperforms the CNN/DAPCA. Is this correct?
Answer: In the example of image analysis (which btw was specifically designed for the DANN paper), DANN does show superior performance. This fact was clearly indicated in the manuscript. The goal of the comparison with DANN was not to outperform it but to see how performant is DANN compared to the proper baseline linear method (and we suggest DAPCA for this purpose). In order to make this point even more clear we introduced, at the end of the Discussion, the following text:
As it was expected, our study shows that neural-network-based classifiers equipped with an adversarial module that tries to distinguish the source from the target domains (such as DANN) achieve better performance than the linear and more constrained DAPCA approach when tested on imaging data. This is partly explained by the fact that the convolutional layers of DANN are trained on the information from both source and target domains, while in our comparison DAPCA used the image representation trained on the source domain only. Linear methods such as DAPCA are deterministic, computationally efficient, reproducible, and relatively easily explainable. Therefore, the linear approaches occupy a niche in those machine learning applications where such features are more important than the maximum possible accuracy. Training neural networks and especially choosing their architectures remains an art, requiring intuition, experience and a lot of computational resources, but this can lead to superior in terms of accuracy results. In a sense, DAPCA stays in the same position to DANN as PCA to the auto-associative neural networks (neural-network-based autoencoders) \cite{Kramer1991}. However, PCA was introduced almost a century before the neural-network-based autoencoders, while a standard fully deterministic and computationally efficient linear approach to domain adaptation based on optimization and using labels from the source domain, is still lacking. Introducing Domain adaptation PCA fills this gap.
Round 2
Reviewer 1 Report
This manuscript has been improved a lot, and it could be neater in terms of writing after a minor revision.
Author Response
We have made another round of extensive grammar checking and corrected several remaining problems. We also revised some of the definitions in Table 1, making them more precise.
Reviewer 2 Report
Thank you for including the notes on CORAL and MMD, and including the image of the feature space.
I was hoping for an actual quantitative comparison, and it looks like you used PCA as the embedding for CORAL? I'm not so sure that this is fair.
Author Response
We added the quantitative comparison between linear DA methods applied to a toy example to Figure 4.
We agree with the Reviewer that in the case of CORAL, we could perform the comparison in the entire feature space, and choosing one particular low-dimensional projection might not give the best quantification results (even though in our specific example, it does not appear to be the case).
From Figure 4, it can now be seen that CORAL correctly aligned the second moments of the source and target distributions, which definitely improved the classifier results. However, due to the simulated important class disbalance in the target dataset, this expectedly does not lead to the good alignment of two data point clouds.
Of note, we double-checked our Python implementation of CORAL by comparing it with the original MATLAB code applied to this example. The MATLAB CORAL code is now provided for reference in the same repository as DAPCA, https://github.com/Mirkes/DAPCA/blob/main/Compute_CORAL.m.
We also observed that the usual L1 normalization makes the results worse. Hence, we used only zscore computation for the preprocessing (the results were similar when only the mean value vectors were subtracted from the source and the target datasets individually).
Reviewer 3 Report
The revised paper is publishable.
Author Response
We thank the reviewer for his positive feedback
Round 3
Reviewer 2 Report
Thank for you adding more results.
On reflection, I feel that publishing this paper would encourage people to use this method, which I don't think is a good idea.
Author Response
We respect the reviewer's feelings and intuition, but we are confident that the use of the suggested domain adaptation method can be beneficial in certain situations. Our confidence is based on several method validations we have performed, as reported in the manuscript. In particular, in a synthetic example, imitating the case when the dominant direction of variance does not match the direction of the best separation of classes and when the target domain has a different class balance compared to the source domain, DAPCA expectedly outperforms other linear domain adaptation methods that usually ignore the class label information in the source domain. This is, for example, a frequent case of the data in bioinformatics (e.g., omics data) where the first principal components are frequently connected to technical or irrelevant biological classification factors. We show that applying DAPCA instead of the standard PCA for dimensionality reduction is a meaningful step in single-cell data integration application. As a matter of fact, DAPCA is already integrated into some of the packages developed for this purpose, such as https://www.biorxiv.org/content/10.1101/2022.11.02.514912v1. In addition, DAPCA provides a simple and fast approach to domain adaptation in the standard example of Amazon reviews and similar settings.
We can not claim, of course, based on the use of only these examples, that DAPCA must be the method of choice in all possible domains. In particular, in the case of image analysis, neural network-based architectures based on adversarial learning show an impressive accuracy of domain adaptation being a more flexible approximator type than a more regular linear approach.
To highlight that the suggested method has its limitations, we added two paragraphs to the discussion section.